# Role of Insulin Resistance as a Mediator of the Relationship between Body Weight, Waist Circumference, and Systolic Blood Pressure in a Pediatric Population

**DOI:** 10.3390/metabo13030327

**Published:** 2023-02-23

**Authors:** Simonetta Genovesi, Laura Montelisciani, Marco Giussani, Giulia Lieti, Ilenia Patti, Antonina Orlando, Laura Antolini, Gianfranco Parati

**Affiliations:** 1School of Medicine and Surgery, University of Milano-Bicocca, Via Cadore 48, 20900 Monza, Italy; 2Cardiologic Unit, IRCCS, Istituto Auxologico Italiano, 20100 Milan, Italy

**Keywords:** BMI, children, HOMA-index, hypertension, mediation analysis, systolic blood pressure, waist circumference

## Abstract

Excess weight and high waist circumference (WC) are associated with increased blood pressure (BP), starting from the pediatric age. Insulin resistance is associated with elevated BP in childhood. The aim of the study was to assess the role of insulin resistance in mediating the relationship between body mass index (BMI), WC, and BP values in a pediatric population referred to a cardio-pediatric center for the presence of one or more cardiovascular risk factors. In 419 children (mean age 10.7 [standard deviation, SD 2.5] years), the following parameters were collected both in basal conditions and after 18.6 (SD 9.3) months of follow-up during which a treatment based on lifestyle and dietary modifications was given: systolic and diastolic BP (SBP and DBP), WC, plasma glucose, and insulin values. The HOMA (Homeostasis Model Assessment)-index was considered as an expression of insulin resistance. At baseline there was a significant correlation between HOMA-index and SBP z-score (β = 0.081, *p* = 0.003), and insulin resistance was a mediator of the relationship between BMI and SBP z-score (*p* = 0.015), and between waist circumference to height (WtHr) and SBP z-score (*p* = 0.008). The effect of BMI z-score modifications on SBP z-score changes from baseline to follow-up was totally mediated by HOMA-index changes (*p* = 0.008), while HOMA-index only partially mediated the effect of WtHr modifications on SBP z-score changes (*p* = 0.060). Our study strongly suggests that, in a pediatric population at cardiovascular risk, the HOMA-index is an important mediator of the relationship between BMI, WC and SBP.

## 1. Introduction

Several studies have shown an association between excess weight and elevated blood pressure (BP) values in children and adolescents [1,2]. Similar data have been observed regarding the relationship between waist circumference (WC) and BP in this age group. Waist circumference improves the ability of body mass index (BMI) to identify hypertension (HT) in obese (OB) children; moreover, WC correlates with BP values in all weight classes [3,4]. However, not all overweight (OW) and OB children and adolescents have high BP values. In addition, a certain fraction of children and adolescents have elevated BP, even without being in excess weight [5,6,7].

It has been hypothesized that in pediatric populations insulin resistance is associated with HT independently of possible confounding factors [8]. It is also known that OW or OB subjects often have higher values of plasma insulin and insulin resistance (expressed as Homeostasis Model Assessment, HOMA-index) than normal-weight peers, even in pediatric age [9].

In previous studies, we described that non-pharmacological treatment based on lifestyle modifications was associated with a reduction in BP, BMI and WC values in a pediatric population with excess weight and/or elevated BP values [10,11].

The question we wanted to address with this study is whether the association between excess weight (and high WC values) and elevated BP is, at least in part, mediated by insulin resistance levels. To this aim, we evaluated the role of the HOMA-index in determining the relationship between BMI, WC, and BP values in a pediatric population referred to a cardio-pediatric center for the presence of one or more cardiovascular risk factors including elevated BP, and/or excess weight, and/or dyslipidemia.

First, we performed a cross-sectional analysis to assess the relationship between systolic (SBP) and diastolic (DBP) BP z-score, BMI z-score, waist-to-height ratio (WtHr) and HOMA-index at recruitment. Next, a longitudinal analysis was conducted to evaluate the relationship between changes in SBP and DBP z-score, BMI z-score and WtHr, and HOMA-index modifications after a period of non-pharmacological treatment based on the correction of dietary and poor lifestyle behaviors.

## 2. Materials and Methods

### 2.1. Subjects

The study population consists of a cohort of 419 children and adolescents who, due to the presence of elevated BP and/or excess weight and/or altered lipid profile, were consecutively referred to the Cardiovascular Risk Assessment in Children Unit from December 2012 to October 2022. The children were referred by family pediatricians who follow children of all socio-demographic conditions free of charge. The Unit is public and access is free, as is the care provided. The following exclusion criteria did not allow enrollment: type 1 and type 2 diabetes (*n* = 2), secondary hypertension of any type (*n* = 6), ongoing antihypertensive drug treatment (*n* = 2). Children for whom it was necessary to start pharmacological treatment to reduce BP values during follow-up (*n* = 22) were excluded from the final analyses. The Unit for Cardiovascular Risk Assessment in Children consisted of the following members: a pediatrician, a nephrologist, a cardiologist, and a nutrition expert.

### 2.2. Baseline and Follow-Up Assessments

In all children, the following parameters were examined at baseline and at the end of the follow-up period: weight, height, WC and BP. Between the baseline and final assessment, a minimum of three to a maximum of six additional periodic visits were made (approximately one every three to four months), during which the same variables were recorded again. At each visit, family members were asked for information to assess adherence to the directions given by the team: change in diet, increase in physical activity, reduction in time spent watching television or playing video games, and the responses were recorded in the participants’ medical records. If a worsening of weight and/or blood pressure was found during a follow-up visit compared with the previous visit, an interview was conducted by the staff (pediatrician, cardiologist, and nutritionist) to ascertain whether and what part of the planned intervention had been disregarded (non-adherence to the diet, lack of physical activity, excess sedentary lifestyle). In particular, a careful dietary history was repeated to understand if the child had particular difficulties in accepting the proposed diet. In this case, modifications and substitutions were suggested, maintaining the total caloric intake and the balance between different macronutrients, but taking into account the patient’s tastes and preferences. It was also checked whether there were any difficulties or errors on the part of the parents in the preparation of meals, in the dosage of foods given, and in the amount of the seasoning. If deemed necessary, more frequent follow-up visits were prescribed. In case of concerns or problems outside of the visits, parents had the option of contacting members of the clinical team (physicians or nutritionists) via e-mail and obtaining the necessary explanations.

### 2.3. Anthropometric Parameters and Blood Pressure 

The anthropometric parameters measured were height, body weight, and WC. The precision applied for data recording was 100 g for body weight and 1 cm for height. Body mass index was calculated as participants’ weight in kilograms divided by the square of height in meters. Body mass index z-scores were calculated using prevention tables from the Centers for Disease and Control prevention charts available at https://www.cdc.gov/growthcharts/clinical_charts.htm (accessed on 9 January 2023). Weight class was defined according to the International Obesity Task Force classification [12] distinguishing between normal weight (NW), OW and OB individuals. Waist circumference was recorded with an accuracy of 1 cm while the study participant was standing. Waist measurement was performed as recommended by Lohman et al. [13,14]. Waist-to-height ratio (WtHr) was obtained by dividing WC by height. Blood pressure was measured with a specific oscillometric device validated for use in children and recommended by the 2016 European Society of Hypertension Pediatric guidelines (Omron 705IT; Omron Co, Kyoto, Japan) [15]; special care was taken to use an appropriately sized cuff. 

Blood pressure values were recorded after a rest period of at least 5 min and while participants were seated. The measurement was taken 3 times (at intervals of a few minutes), and the mean value of the second and third measurements was recorded. The percentiles and z-scores of SBP and DBP were calculated based on nomograms from the National High Blood Pressure Education Program (NHBPEP) Working Group on High Blood Pressure in Children and Adolescents [13,14]. Children were classified according to the mean of the two measurements as follows: normotensive (NT) if the percentiles of SBP and DBP were both <90th; high normal (HN) if the percentiles of SBP and/or DBP were ≥90th, but both <95th; hypertensive (HT) if the percentiles of SBP and/or DBP were ≥95th.

### 2.4. Biochemical Parameters

Blood samples were taken after a 12-h fasting period to measure serum concentrations of total cholesterol, high-density lipoprotein (HDL), triglycerides, glucose, and insulin. Commercial kits, normally used for routine patient examinations, were used for all analyses. [Cobas Roche colorimetric enzymatic cholesterol Gen.2 test, for total cholesterol assay; homogeneous-phase colorimetric enzymatic test HDL cholesterol Gen.4 Cobas Roche, for HDL cholesterol; colorimetric enzymatic test Triglycerides Cobas Roche, for triglyceride assay; hexokinase enzymatic method Glucose HK Gen.3 Cobas Roche, for glucose assay; immunoassay in ElectroChemiLuminescence Elecsys Insulin Cobas Roche, for insulin assay]. The HOMA index was calculated by dividing the product of serum insulin (µU/mL) and serum glucose (mmol/L) by 22.5 [16].

### 2.5. Recommended Lifestyle Modifications 

All participants were encouraged to engage in at least two to three hours of structured physical activity per week [17], to engage in more spontaneous unstructured physical activity, and to reduce sedentary activities such as playing video games or watching TV to a maximum of one hour per day, as recommended by the Italian Society of Pediatrics (https://sip.it/2017/09/25/non-solo-sport-ma-anche-gioco-la-piramide-dellattivita-fisica-e-motoria-per-combattere-lobesita/, accessed on 21 February 2023). Children were encouraged to choose sports of their liking for an appropriate number of hours per week.

All participants were given general advice on how to achieve a healthy and balanced diet (more fruits and vegetables, low-fat dairy products, lower intake of free sugars and elimination of soft drinks) with proper salt intake (a maximum of 5 g per day near equivalent to 2 g of sodium) following the World Health Organization (WHO) guideline) (www.who.int/data/gho/indicator-matadata-registry/imr-details/3082, accessed on 21 February 2023)

At the baseline visit, parents of the children and adolescents were interviewed by an experienced nutritionist to assess the eating habits and physical activity levels of the participants. Based on the information obtained, appropriate and individualized changes in lifestyle and nutritional habits were proposed to each participant. Once it was determined what the appropriate caloric intake was and how it should be divided into protein, glycides, and lipids in the dietary pattern, specific interviews were conducted between the child, parents, and nutritionist to perform a diet that took into account the individual child’s preferences and the families’ needs. A personalized dietary scheme was prepared for all participants with the help of specially designed software (Dietosystem, Ds Medica, Milan, Italy).

Depending on each participant’s actual body weight and blood pressure status (excess weight isolated, or high, or a combination of both risk factors), additional specific recommendations were provided.

Excess weight. Overweight and obese subjects were subjected to a weekly dietary program, the calorie content of which had been previously calculated based on the Schofield equation [18], which considers basal metabolic rate, and based on functional metabolism [19]. Young children were asked to follow a balanced normocaloric regimen that matched the estimated energy expenditure, while adolescents with severe excess weight were recommended to follow a mildly hypocaloric (−10%) diet. The dietary-behavioral treatment implemented to reduce HOMA-index values coincided with that to reduce excess weight. Qualitatively, consumption of non-starchy fruits and vegetables rich in fiber, vitamins and minerals, citrus fruits, legumes and preferably whole grains, lean meats, fresh cheeses, fish and nuts, and unsweetened dairy products was encouraged. The intake of sugary, carbonated and soft drinks was excluded. Fruit juices, starchy vegetables such as potatoes, squash, and corn, processed snacks and canned foods, sweets, ice cream, and chocolate were limited.

Elevated blood pressure. In HN and HT individuals it was proposed to reduce salt intake to less than 5 g per day, following the WHO guidelines. 

The study protocol was approved by the Local Ethics Committee (Istituto Auxologico, Milano, Italy) (RICARPE 2015_10_20_02) and informed consent was obtained from the children’s parents. 

### 2.6. Statistical Analysis

Continuous variables were expressed as mean and standard deviation and categorical variables by count and relative frequency (%) of each category.

The raw data of BMI and SBP and DBP were converted to z-score. The BMI and SBP z-scores are a function of age and sex, according to the formula:Z-score = [(X/M)^L^ − 1]/(L*S)
were L is the Box-Cox transformation, M is the median, and S is the generalized coefficient of variation [20].

Univariate analyses to compare the characteristics of the children at baseline and at follow-up were conducted through the *t*-test for paired data and by the McNemar test for categorical variables.

The relationship between the variables under the baseline condition was investigated through the linear regression coefficient and graphically by scatter plot with regression line.

To investigate the effect of the time between the baseline time and the follow-up the deltas of each variable of interest have been calculated: all the deltas used in the analysis were obtained as the difference between the baseline value and the follow-up value of each variable. At the follow-up time, univariate linear regression models explored the association between the outcome, i.e., delta SBP z-score, and delta of explanatory variables.

On the basis of the significance of the regression coefficients at baseline, a mediation analysis was performed to deeply understand the role of the HOMA-index as possible mediator between BMI z-score or WtHr on the SBP z-score. The mediation analysis was performed to evaluate the role of HOMA-index in mediating the relationship between BMI z-score, WtHr z-score, and BP z-score values under baseline conditions and the effect of changes in BMI z-score and WtHr on changes in BP z-scores after the follow-up period.

In the mediation analysis, a third variable (called the mediator) is added to the analysis of the relationship between the independent and dependent variables in order to improve understanding of this relation. A mediator is a variable that transmits the effect of the exposure on an outcome. The goal of this analysis is to partition the total treatment effect into two components: the indirect effect that occurs due to the mediator and the direct effect that captures the treatment effect, net of the mediator. Perfect mediation occurs when the relationship between a treatment and the outcome has been completely explained by the mediator. In addition to the mediation analysis at baseline, another mediation model was applied to verify the mediation role of the delta HOMA-index between the delta BMI z-score or delta WtHr on the delta SBP z-score.

## 3. Results

Table 1 describes the characteristics of the study population under baseline conditions and at the end of follow-up [18.6 (standard deviation, SD 9.3) months]. At recruitment, the mean age was 10.7 (SD 2.5) years. Fifty-seven percent of the children were male and 45.4% had begun pubertal development. The percentage of NW individuals was 17.9%, OW were 34.4% and OB were 47.7%. The percentages of study participants with normal BP, HN and HT were 56.6%, 14.5% and 28.9%, respectively.

HOMA-index values were significantly higher in children with SBP and/or DBP values ≥ 90th percentile (HN + HT) compared with those with values < 90th percentile [2.61 (1.9) vs. 3.03 (2.0), *p* = 0.030]. Similar results were observed regarding subjects with SBP z-score values ≥ 90th percentile in comparison with those with values < 90th percentile [2.63 (1.9) vs. 3.04 (2.7), *p* = 0.038]. In contrast, there was no difference between the HOMA-index values of children with DBP z-score ≥ 90th percentile compared with those with DBP z-score <90th percentile [2.73 (1.9) vs. 3.18 (2.0) *p* = 0.092].

HOMA-index values were significantly associated with both BMI z-score (β = 0.151, *p* < 0.001) and WtHr value (β = 1.250, *p* < 0.001). There was also a significant association between HOMA-index and SBP z-score (β = 0.081, *p* = 0.003), but not between HOMA-index and DBP z-score (β = 0.030, *p* = 0.123). The association between HOMA-index and SBP z-score was significant both in children with excess weight (β = 0.066, *p* = 0.019) and in those with normal weight (β = 0.391, *p* = 0.012) (Figure 1). BMI z-score was not significantly related to either SBP (β = 0.088, *p* = 0.128) or DBP (β = 0.050, *p* = 0.163) z-score. Waist to height ratio correlated significantly with DBP z-score (β = 0.0100.021, *p* = 0.020), but not with SBP z-score (β = 0.010, *p* = 0.196).

The red lines delineate the distribution of variables in NW individuals and the blue lines delineate the distribution of variables in excess weight (OW + OB) individuals. The red dots in the scatter plots represent NW individuals and the blue dots represent individuals with excess weight (OW + OB).

Multivariable regression models showed that HOMA-index remained significantly associated with SBP z-score after adjusting for BMI z-score or WtHr, while both BMI z-score and WtHr were not significantly related with SBP z-score. There was no significant association between HOMA-index and DBP z-score (Table 2).

Mediation analysis showed that, under baseline conditions, a significant indirect effect was present between BMI z-score and SBP z-score, and that this association was completely mediated by HOMA-index values (indirect effect = 0.050, *p* = 0.015) (Figure 2a). No significant indirect effect was present between BMI z-score and DBP z-score (indirect effect = 0.020, *p* = 0.257) (Appendix A). Similar results were observed regarding WtHr: an indirect effect was present between WtHr and SBP z-score, mediated by HOMA-index (indirect effect = 0.008, *p* = 0.008) (Figure 2b), but not between WtHr and DBP z-score (indirect effect = 0.002, *p* = 0.254) (Appendix A). The effect of BMI z-score and WtHr on SBP z-score net of mediator (direct effect) was not significant.

At the end of the follow-up, the percentage of children with excess weight (OW + OB) decreased from 82.1% to 69.7% and that of subjects with elevated BP values (HN + HT) from 43.4 to 23.9% (*p* < 0.001, Table 1). At follow-up, the percentage of study participants with normal weight was 30.3%, that of OW 42.7% and that of OB 27.0%. Children with normal BP values were 71.1%, those HN 11.2% and those HT 17.7%. HOMA-index mean went from 2.80 to 2.72 (*p* = 0.654). There was a significant association between delta BMI z-score and delta SBP z-score (β = 0.26, *p* = 0.025), between delta WtHr and delta SBP z-score (β = 0.03, *p* = 0.002) and between delta HOMA-index and delta SBP z-score (β = 0.08, *p* = 0.001) from baseline to follow-up (Table 3).

The association between delta of HOMA-index and delta of SBP z-score was still present after adjustment for BMI z-score, WtHr and transition from pre-puberty to puberty. While the significant relationship between delta BMI z-score and delta SBP z-score disappeared in the multivariable model, delta WtHr remained an independent predictor of delta SBP z-score from baseline to follow-up (Table 4).

To test the robustness of HOMA-index mediation in the relationship between BMI, WtHr, and SBP observed under basal conditions, we analyzed whether this mediation was confirmed when we assessed the association between deltas of BMI and WtHr from baseline to follow-up and deltas of SPB z-score. Figure 3 displays this analysis. Mediation analysis showed that the effect of BMI z-score changes on SBP z-score changes was totally mediated by HOMA-index modifications (indirect effect = 0.11, *p* = 0.008, Figure 3a). Changes in HOMA-index only partially mediated the effect of modifications of WtHr on changes in SBP z-score (indirect effect = 0.005, *p* = 0.060, Figure 3b). A significant direct effect of delta WtHr on delta SBP z-score was present (*p* = 0.011). The results of the mediation analysis were similar after adjustment for the transition from pre-puberty to puberty, although the mediating role of the HOMA-index was slightly weakened (Appendix A). Mediation analysis showed that the effect of deltas BMI z-score and deltas WtHr on deltas DBP z-score from baseline to follow-up was not mediated by HOMA-index (Appendix A).

## 4. Discussion

Our study in addition to confirming the association between insulin resistance and blood pressure in children, strongly suggests that, in a pediatric population at cardiovascular risk, the HOMA-index is an important mediator of the relationship between BMI and waist circumference with systolic blood pressure. This evidence is proved not only under baseline conditions, but also after a period when children were undergoing an intervention based on a non-pharmacological approach. Changes in systolic blood pressure following changes in BMI and waist circumference at the end of follow-up were significantly mediated by simultaneous modifications of HOMA-index. 

Several studies demonstrated an association between arterial hypertension and insulin resistance and this is also a problem for pediatric patients [21]. Insulin resistance is involved in the development of hypertension through various mechanisms. Hyperinsulinemia contributes directly to the development of endothelial dysfunction by inhibiting the production of nitric oxide, a potent vasodilator, thereby promoting increased vascular resistances [22] and an association between increased HOMA-index and impaired endothelial function since childhood has been proven [23]. Furthermore, insulin resistance leads to stimulation of the sympathetic nervous system resulting in vasoconstriction and increased blood pressure [24]. Activation of the sympathetic nervous system also results in an activation of the renin-angiotensin system with a consequent increase in angiotensin II levels and an increment not only in peripheral resistances, but also in renal sodium reabsorption [25]. The HOMA index is generally accepted as a measure of insulin resistance, however, a number of other indices have recently been suggested as an expression of this clinical condition [26]. It has been suggested that one of these, the triglyceride-glucose (TyG) index was superior to the HOMA index in the prediction of hypertension in adults [27]. Data on the pediatric population are very few, however, a Mexican study showed that the elevated TyG index is significantly associated with the presence of prehypertension and hypertension in children and adolescents. Future studies confirming this finding may certainly be useful in understanding the role of insulin resistance and the genesis of hypertension in children [28].

It is well known that obesity is a major cause of insulin resistance [17,29] and it has been shown that an association between insulin resistance and hypertension is already present in both overweight and obese children and adolescents [30,31]. Interestingly, this relationship is also evident in normal weight children: in a pediatric population, in which all weight classes were represented, the HOMA-index was shown to be an independent predictor of hypertension, after adjustment for body weight and fat distribution. In addition, HOMA-index was independently associated with absolute z-scores of blood pressure [8]. Despite the known association between excess weight and high blood pressure, not all overweight and obese children and adolescents are hypertensive. In a large cohort (*n* = 1201) of obese children and teenagers, only 25.9% had systolic blood pressure values ≥ 90th percentile. However, individuals with the “metabolically unhealthy obesity” phenotype, in addition to having higher blood pressure than their metabolically healthy peers, also had significantly higher levels of HOMA-index [32]. These results were later confirmed in a smaller study [33].

In our study population, under basal conditions, there was a significant association between HOMA-index and SBP z-score values, whereas this finding was not present when we consider the relationship between BMI and waist circumference with blood pressure. Moreover, HOMA-index was higher in the sub-group of children with SBP greater than 90th percentile and the association between HOMA-index and systolic blood pressure was strongly significant even after adjustment for indexed values of BMI and waist circumference. Children with higher blood pressure values were those with higher level of insulin resistance. It is interesting that this applied to children with excess weight as well as those with normal weight. As expected, HOMA-index values were higher in children with higher BMI z-scores, however, the relationship between HOMA-index and SBP z-score was present regardless of weight class. These observations suggested to us that the degree of insulin resistance might be a mediator between body weight, waist circumference, and blood pressure values. The mediation analysis confirmed that that HOMA-index plays an important role in mediating the effect of both BMI z-score and WtHr on systolic blood pressure.

To validate the results derived from the cross-sectional analysis, we wanted to test whether the association between HOMA-index and systolic blood pressure was maintained when blood pressure values changed. Since in our population the changes in blood pressure at follow-up were accompanied by consensual changes in weight and waist circumference, we also assessed whether the HOMA-index maintained a mediating role in this relationship. Changes in weight and waist circumference at follow-up, were significantly associated with changes in systolic blood pressure, as were changes in HOMA-index. Again, deltas of HOMA-index were associated with blood pressure deltas independently of deltas of BMI and waist circumference. HOMA-index was the main mediator of the effect of BMI on blood pressure modifications. On average, the intervention was associated with a reduction in BMI in our sample. A randomized study showed beneficial effects on body weight associated with insulin resistance improvement in overweight children undergoing dietary treatment [34]. Moreover, in our previous study, we observed that the factor most strongly associated with improved insulin resistance values in a pediatric population undergoing a dietary-behavioral intervention was the decrease in BMI z-score [11]. In the present study, the greater the weight reduction during follow-up, the greater the decrease in systolic blood pressure, and this effect appeared totally mediated by the amount of reduction in insulin resistance. While mean BMI z-score values decreased significantly from baseline to follow-up, the same was not true for HOMA-index values. Since we observed a significant association between changes in HOMA-index and changes in SBP z-score, this means that children in whom insulin resistance increases may experience an increase in systolic blood pressure. Taken together, these findings support the hypothesis that HOMA-index is an important mediator of the effect of BMI on systolic blood pressure.

More complex seems to be the relationship between central adiposity, HOMA-index and systolic blood pressure. As with BMI, reductions in waist circumference were also significantly associated with reductions in systolic blood pressure at the end of follow-up, however, this effect seemed to be only partly mediated by insulin resistance modifications. It is reasonable to think that visceral fat has close linkages with other factors and mechanisms associated with blood pressure values. For example, reduced adiponectin levels are associated with both elevated waist circumference and high blood pressure values in adults and children [35,36]. An increase in adiponectin levels associated with waist circumference reduction could have a direct effect on blood pressure values in our population. It has been suggested that a number of novel adipokines may be involved in the pathogenesis of cardiovascular disease related to excess weight [37]. In addition, it was shown that serum uric acid increases as the waist circumference to height ratio increases [38], and a relationship between hyperuricemia and elevated blood pressure values has been shown as early as in pediatric age [38,39]. Recently it has been suggested that insulin resistance may be a mediator of the effect of serum uric acid in leading to increased vascular stiffness [40,41]. A Chinese study performed in adults, showed that elevated serum uric acid levels were associated with an increased risk of incident hypertension, and insulin resistance played a mediating role in the relationship between serum uric acid and hypertension [42]. It is possible to speculate that visceral fat, of which waist circumference is an expression, may exert its effects on blood pressure independently of insulin resistance or that newly discovered cytokines may in turn mediate between insulin resistance and its effects on blood pressure. It is therefore possible to think that there is a complex interaction between central adiposity, hyperuricemia, insulin resistance, and blood pressure even in our population. 

Various indices of relative weight have been proposed and applied to indicate obesity or body fatness [43]. A child’s body is a growing organism, for this reason it’s incorrect to use the raw data of BMI as an indicator of body mass. Among the indices proposed to evaluate adiposity in children, we have chosen the BMI z-score, because it is the most widely used index in all available pediatric studies. In this way our data can be compared with those of other authors. Furthermore, the most recent Clinical Practice Guideline for obesity evaluation and treatment of the American Academy of Pediatrics recommends using the BMI z-score for studies in children and adolescents, in particular for assessing longitudinal change in adiposity over time, as in the case of our study [44]. Several limitations of this study need to be acknowledged. First, the study is not a randomized trial, so a control group is missing. A control group was not included because in our opinion (as well as in the opinion of the Ethics Committee) it would have been unethical not to offer any kind of treatment to hypertensive and/or overweight children referred to our center by their family pediatricians because of a clinical problem. Second, we have no evidence of patients’ compliance with the intervention, particularly with regard to the low-sodium diet adherence. The effect of dietary salt restriction as a tool to reduce blood pressure is known and documented in adults [45] and suggested for pediatric populations [46]. However, performing a 24-h urine sodium collection before and after our intervention period would not have been enough to give a measure of the actual dietary sodium intake of our patients, but would only have given information on the intake of salt the day preceding the test. To have consistent data on the level of adherence to the low sodium diet, we would need to measure urine sodium repeatedly and frequently throughout the follow-up period. Unfortunately, this was not possible, given the size of the sample and the need not to ask too much of the children’s families. It should also be emphasized that sodium intake estimation formulas derived from urinary sodium are not validated in children. Sodium intake is significantly associated with insulin resistance [47]. It has recently been shown that in excess weight insulin-resistant subjects reduction of dietary sodium improves cardiac function, and this effect may be associated with improvement in insulin resistance [48]. We can only speculate, without having any evidence that hypertensive children with the greatest reductions in blood pressure at follow-up were those in whom the low-sodium diet was most closely followed. Third, mediation analysis only allows to say that a factor acts as a mediator on the effect of one variable on another variable, but it does not prove a causal relationship. Furthermore, as to our knowledge no studies on this topic in pediatric populations using this statistical approach are available, our data need confirmation.

In conclusion, our study shows that insulin resistance plays an important role in determining systolic blood pressure values in children. For the same BMI values and waist circumference, individuals with higher HOMA-index levels would therefore be at higher risk of hypertension. Furthermore, the reduction in blood pressure values that comes with a decrease in body weight is closely associated with a concomitant decrease in insulin resistance. Children in whom improved dietary and lifestyle habits lead to blood pressure values improvement would be those in whom the degree of insulin resistance is most reduced. Our data suggest the importance of insulin dosing in children and adolescents with excess weight and/or high blood pressure and that reducing insulin resistance may be a potentially valuable strategy in lowering high blood pressure in this population. Children represent a unique model for studying the pathophysiology of essential hypertension, as confounding factors such as aging, comorbidities, medications and smoking are absent. Thus, our data could represent a noteworthy piece in the complex puzzle of the etiopathogenesis of essential hypertension.

## Figures and Tables

**Figure 1 metabolites-13-00327-f001:**
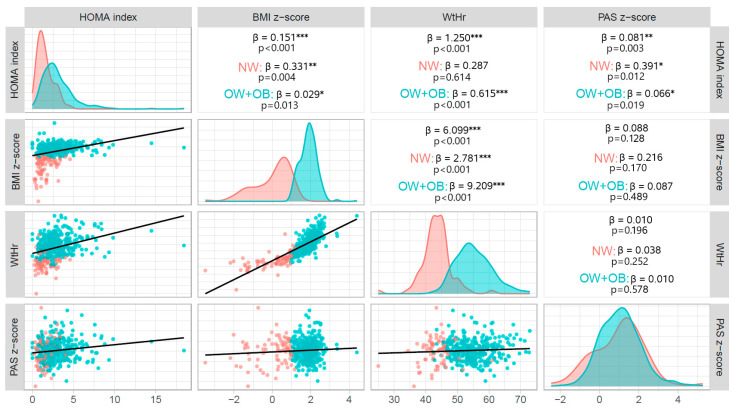
Correlations between BMI z-score, WtHr, HOMA-index and SBP z-score at baseline. BMI: body mass index; NW: normal weight; OB: obese; OW: overweight; HOMA: Homeostasis Model Assessment, SBP: systolic blood pressure; WtHr (%): waist to height ratio; * *p* < 0.5; ** *p* < 0.01, *** *p* < 0.001.

**Figure 2 metabolites-13-00327-f002:**
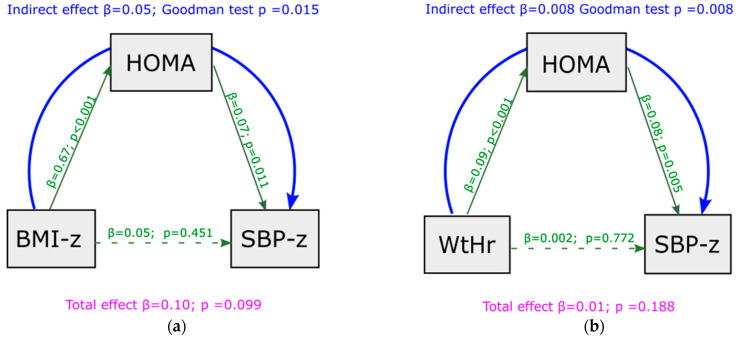
Mediation analysis model with SBP z-score as outcome at baseline, including HOMA-index as a mediator of the effect of BMI z-score (panel (**a**)) and WtHr (panel (**b**)) on SBP z-score. BMI: body mass index; HOMA: Homeostasis Model Assessment; SBP: systolic blood pressure; WtHr: waist to height ratio.

**Figure 3 metabolites-13-00327-f003:**
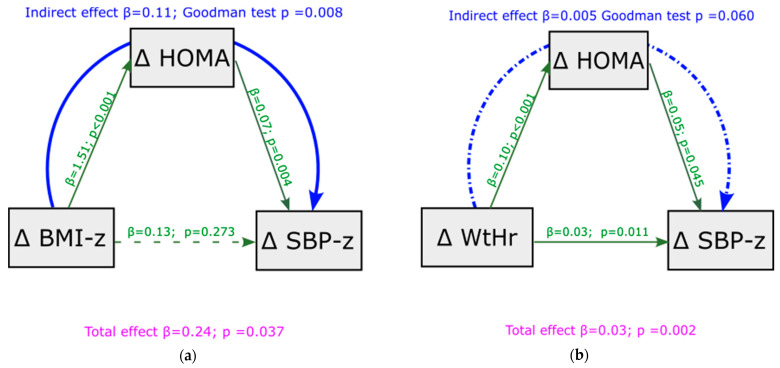
Mediation analysis model with delta SBP z-score from baseline to follow-up as outcome, including delta HOMA-index as a mediator of the effect of delta BMI z-score (panel (**a**)) and delta WtHr (panel (**b**)) on delta SBP z-score. BMI: body mass index; HOMA: Homeostasis Model Assessment; SBP: systolic blood pressure; WtHr: waist to height ratio.

**Table 1 metabolites-13-00327-t001:** Characteristics of the study population under baseline conditions and at the end of follow-up.

Parameter	Baseline (*n* = 419)	Follow-Up (*n* = 419)	*p*-Value
Age (years), mean (SD)	10.70 (2.46)	12.21 (2.48)	<0.001
Gender (males), *n* (%)	239 (57.04)	-	-
Puberty, *n* (%)	190 (45.35)	313 (74.70)	<0.001
Weight (kg), mean (SD)	52.67 (18.34)	57.28 (18.12)	<0.001
Height (m), mean (SD)	1.46 (0.15)	1.54 (0.14)	<0.001
BMI (kg/m^2^), mean (SD)	24.04 (4.76)	23.70 (4.64)	0.004
BMI (z-score), mean (SD)	1.54 (0.92)	1.24 (0.93)	<0.001
Excess weight (OW + OB), *n* (%)	344 (82.10)	292 (69.69)	<0.001
Waist circumference (cm), mean (SD)	77.41 (12.94)	77.19 (12.00)	0.090
WtHr (%), mean (SD)	53.10 (7.10)	50.22 (6.62)	<0.001
SBP (mmHg), mean (SD)	116.01 (13.86)	115.00 (13.03)	0.039
SBP (z-score), mean (SD)	1.06 (1.08)	0.70 (1.02)	<0.001
DBP (mmHg), mean (SD)	67.62 (8.55)	66.68 (8.21)	0.038
DBP (z-score), mean (SD)	0.53 (0.72)	0.33 (0.69)	<0.001
SBP and/or DBP ≥ 90th percentile, *n* (%)	182 (43.44)	121 (28.88)	<0.001
SBP ≥ 90th percentile, *n* (%)	170 (40.57)	116 (27.68)	<0.001
DBP ≥ 90th percentile, *n* (%)	64 (15.27)	36 (8.59)	0.001
Glucose (mg/dL), mean (SD)	83.56 (7.14)	84.39 (7.68)	0.044
Insulin (µU//mL), mean (SD)	13.46 (8.66)	13.05 (9.11)	0.357
HOMA-Index, mean (SD)	2.80 (1.94)	2.72 (2.06)	0.654
Total cholesterol (mg/dL), mean (SD)	97.17 (28.36)	92.07 (27.14)	<0.001
HDL cholesterol (mg/dL), mean (SD)	54.00 (12.47)	54.16 (12.38)	0.500
Triglycerides (mg/dL), mean (SD)	76.24 (38.33)	74.02 (35.95)	0.249

BMI: body mass index; DBP: diastolic blood pressure; HDL: high density lipoprotein; HOMA: Homeostasis Model Assessment; OB: obese; OW: overweight; SBP: systolic blood pressure; SD: standard deviation; WtHr: waist to height ratio.

**Table 2 metabolites-13-00327-t002:** Association between HOMA-index with Systolic Blood Pressure z-score and Diastolic Blood Pressure z-score at univariate (Model a) and multivariable (Model b) regression model. BMI: Body Mass Index; WtHr: Waist to Height ratio; HOMA: Homeostasis Model Assessment; SBP: Systolic Blood Pressure; DBP: diastolic blood pressure.

		SBP z-Score			
		Coefficient	Std. Err.	*p*-Value	[95% Conf. Interval]
Model a	HOMA index	0.08	0.03	0.003	(0.027–0.135)
Model b	HOMA index	0.07	0.03	0.011	(0.017–0.131)
	BMI z-score	0.05	0.06	0.451	(−0.074–0.165)
Model b	HOMA index	0.08	0.23	0.005	(0.026–0.142)
	WtHr	0.002	0.01	0.772	(−0.013–0.018)
		DBP z-score			
		Coefficient	Std. Err.	*p*-value	[95% conf. Interval]
Model a	HOMA index	0.03	0.02	0.123	(−0.008–0.064)
Model b	HOMA index	0.02	0.02	0.256	(−0.016–0.059)
	BMI z-score	0.04	0.04	0.306	(−0.038–0.120)
Model b	HOMA index	0.02	0.02	0.253	(−0.016–0.060)
	WtHr	0.01	0.01	0.072	(−0.001–0.020)

**Table 3 metabolites-13-00327-t003:** Association between deltas of BMI z-score, WtHr and HOMA-index and delta SBP z-score from baseline to follow-up.

	Coefficient	Std. Err.	*p*-Value	[95% Conf. Interval]
Delta BMI (z-Score)	0.26	0.11	0.025	(0.03–0.48)
Delta WtHr	0.03	0.01	0.002	(0.01–0.06)
Delta HOMA-index	0.08	0.02	0.001	0.03–0.13

BMI: body mass index; HOMA: Homeostasis Model Assessment; SBP: systolic blood pressure; WtHr: waist to height ratio.

**Table 4 metabolites-13-00327-t004:** Association between delta of HOMA index and delta SBP z-score from baseline to follow-up at multivariate regression models.

	Coefficient	Std. Err.	*p*-Value	[95% Conf. Interval]
Delta HOMA Index	0.07	0.02	0.004	(0.023–0.121)
Delta BMI (z-score)	0.13	0.12	0.273	(−0.104–0.367)
Delta HOMA Index	0.05	0.03	0.045	(0.001–0.103)
Delta WtHr	0.03	0.01	0.011	(0.007–0.050)
Delta HOMA Index	0.07	0.03	0.017	(0.012–0.121)
Delta BMI (z-score)	0.17	0.13	0.193	(−0.090–0.424)
Pre-puberty to Puberty	−0.15	0.12	0.221	(−0.395–0.092)
Delta HOMA Index	0.05	0.03	0.064	(−0.003–0.106)
Delta WtHr	0.03	0.01	0.029	(0.003–0.048)
Pre-puberty to Puberty	−0.17	0.13	0.179	(−0.424–0.080)

BMI: Body Mass Index; WtHr: Waist to Height ratio; HOMA: Homeostasis Model Assessment.

## Data Availability

Data are available are deposited at the repository that can be reached at the following link: https://zenodo.org/, accessed on 20 December 2022.

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
