# Peer review of "Role of Insulin Resistance as a Mediator of the Relationship between Body Weight, Waist Circumference, and Systolic Blood Pressure in a Pediatric Population"

_metabolites, 2023, doi:10.3390/metabo13030327_

Round 1

Reviewer 1 Report

To evaluate the effect of blood pressure treatment on insulin resistance, this author conducted a 18.6 months follow up study of 419children. From this study, this author strongly suggests that, in a pediatric population at cardiovascular risk, the HOMA-index is an important mediator of the relationship between BMI, WC and SBP. I thought this manuscript contains many critical issues that should be solved. I commented those issues.

In abstract section.

1)  There is no description about the age of target population.

2) Spell out the “HOMA”.

3)  To evaluate the weight and height valance in children, generally there are three indexes. Those are Kaup index, laurel Index, and body mass index. Since which index should be used in epidemiological study are depending on the stage of growth, the target study population should be narrow aged. However, there is no description about the age of target population. And there is no description that explain “BMI” is appropriate index to be used in this study.

4)  Without describing the meaning of “r” this author used this abbreviation. The “r” might be simple correlation coefficient.

5)  Generally, the magnitude of simple correlation coefficient value under |0.2| regards too small to conclude as the clinical significance. Therefore, those values could be no significant after adjusted for potential confounder. Then the correlation between HOMA-index and SBP (r=0.145, p=0.003) could not became a main finding of this study. Multivariable model which adjusted for known confounding factors should be used.

6) I could not understand the meaning of “insulin resistance was a mediator of the relationship between SBP and BMI (p=0.015), and SBP and WC (p=0.008)”. Is this means that the continuous variable of HOMA-index shows significant effect on the association between SBP and BMI and the association between SBP and WC?

Introduction

6) The sentences that shown in last paragraph of this section should be move to statistical analysis section.

7) To under stand the content of this study, this author should describe the hypothesis of this study.

8) Spell out the “HOMA” in this section.

Material and method section

9) The range of age of present study population should be shown. In general clinical practice, BMI is used for participants with aged at and above 16 years while laurel Index is used for participants with aged between 5 to 15 years. Then range of this study should be taken into consideration.

10) The specific number of participants who are excluded from present study should be shown. The number of participants who has diabetes, secondary hypertension, taking antihypertension medication, and the participants who starts to taking antihypertensive medication during follow-up period should be clarified.

11) Multi-adjusted model which adjusted for known confounders should be added in present results. Especially for the influence of sex and age should be taken into consideration.  

Results section

12) As shown in Table 2, the correlation between SBP z-score and delta BMI (z-score) and the correlation between SBP z-score and delta HOMA index, showed wide range of 95% confidence intervals. Therefore, even p-value showed significant value, such results could indicate that the sample size of this population is too small to validate present results. In addition to that, this analyzes were performed by using crude model. Then validity of those results is questionable. At least, the results of good for fit test should be shown.

13) The abbreviation of HOMA should be explain in foot note.

Discussion section

14) By using the multivariable model, the analyzes should be reperformed. And the content of discussion should be changed according to revised results.

Author Response

To evaluate the effect of blood pressure treatment on insulin resistance, this author conducted a 18.6 months follow up study of 419children. From this study, this author strongly suggests that, in a pediatric population at cardiovascular risk, the HOMA-index is an important mediator of the relationship between BMI, WC and SBP. I thought this manuscript contains many critical issues that should be solved. I commented those issues.

We thank the reviewer for his/ her comments and requests for elucidations that allowed us to improve and clarify our manuscript.

In abstract section.

  • There is no description about the age of target population.

The age of the study population is already reported in Table 1 (10.70 [SD 2.46] years at Baseline and 12.21 [SD 2.48] years at Follow-up] and in the Results section. We have also added this information to the abstract as requested. 

  • Spell out the “HOMA”.

The term HOMA was explained in full: Homeostasis Model Assessment

  • To evaluate the weight and height valance in children, generally there are three indexes. Those are Kaup index, laurel Index, and body mass index. Since which index should be used in epidemiological study are depending on the stage of growth, the target study population should be narrow aged. However, there is no description about the age of target population. And there is no description that explain “BMI” is appropriate index to be used in this study.

We thank the reviewer for this comment that allows us to clarify how we performed our analysis. We completely agree with the reviewer that BMI and blood pressure raw values would not be an appropriate measure for children and adolescents. In fact, we have not used them, but used the BMI and blood pressure z-scores. A z-score describes the position of a raw score in terms of its distance from the mean, when measured in standard deviation units. The z-score is positive if the value lies above the mean, and negative if it lies below the mean. The z-score is age- and sex-specific, and is used instead of the raw data in order to compare individuals of different sex and age with respect to a specific parameter (in our case BMI and blood pressure). The BMI z-score and SBP z-score are a function of age, and sex. This is the formula:

Z-score = [(X/M)L - 1]/(L*S)

The LMS parameters are the power in the Box-Cox transformation (L), the median (M), and the generalized coefficient of variation (S). Given these parameters, any desired percentile (for sex and age) can be calculated. (reference: “Construction of LMS Parameters for the Centers for Disease Control and Prevention 2000 Growth Charts – Flegal et al. Natl Health Stat Report. 2013 Feb 11;(63):1-3es”). For this reason, z-score values (both BMI z-score and SBP z-score) are already adjusted by sex and age. Thus, the association between exposure and the outcome is also adjusted for both sex and age. Further adjustment for these two parameters would result in over-correction. An explanation of the meaning of z-scores has been included in the methods section.

Since nomograms obtained from a large enough pediatric population to create z-scores were not available for waist circumference, waist circumference was indexed for height (WtHr), as suggested by several authors (Maffeis et al. Waist-to-Height Ratio: a useful index to identify high metabolic risk in overweight children. J Pediatr 2008;152:207-213; Garnett et al. Waist-to-Height Ratio, a simple option for determining excess central adiposity in youmg people. Int J Obes 2008; 32:1028–1030; Mokha JS et al. Utility of Waist-to-Height ratio in assessing the status of central obesity and related cardiometabolic risk profile among normal weight and overweight/obese children: The Bogalusa Heart Study.BMC Pediatrics 2010; 10:73).

  • Without describing the meaning of “r” this author used this abbreviation. The “r” might be simple correlation coefficient.
  • Generally, the magnitude of simple correlation coefficient value under |0.2| regards too small to conclude as the clinical significance. Therefore, those values could be no significant after adjusted for potential confounder. Then the correlation between HOMA-index and SBP (r=0.145, p=0.003) could not became a main finding of this study.

Linear correlations were replaced by regression models (simple and multi-variable). For this reason, the Pearson correlation coefficients (r) in Figure 1 were replaced by univariate regression coefficients (β).

  • Multivariable model which adjusted for known confounding factors should be used.

As requested by the reviewer, to better evaluate the relationship between HOMA index and systolic blood pressure, we performed two multivariable regression models (one adjusted by BMI z-score and one by WtHr) with SBP z-score as the outcome. The models showed that both BMI z-score and WtHr were not significantly related to SBP z-score, while the effect of HOMA index remained significant after adjusting for BMI z-score or WtHr. This result suggested to us the idea that HOMA could mediate the relationship between BMI (and WtHr) and systolic blood pressure (Table 2). We also added a multivariable model with delta of SBP z-score as the outcome, to evaluate data at follow-up, adjusted for delta of BMI z-score, delta of WtHr and transition from pre-puberty to puberty (Table 4).

  • I could not understand the meaning of “insulinresistance was a mediator of the relationship between SBP and BMI (p=0.015), and SBP and WC (p=0.008)”. Is this means that the continuous variable of HOMA-index shows significant effect on the association between SBP and BMI and the association between SBP and WC?

We thank the reviewer for the question that allows us to clarify the meaning of the analysis we performed. Mediation analysis is a different type of analysis from the classically used analysis (regression analysis) that demonstrate an association between two variables. Mediation analysis reveals whether or not the association between two variables is mediated by another factor and how strong this mediation is. In the mediation analysis, one variable affects a second variable that, in turn, affects a third variable. This last one variable is the mediator. It "mediates" the relationship between a predictor and an outcome. With the sentence “insulin resistance was a mediator of the relationship between SBP and BMI (p=0.015), and SBP and WC (p=0.008)”, we want to mean that predictors (WC and BMI z-score) had a statistically significant effect on outcomes (SBP z-score) through the mediator (HOMA Index) values. These concepts have been better explained in the revised manuscript.

Introduction

  • The sentences that shown in last paragraph of this section should be move to statistical analysis section

As requested, the two sentences have been moved to the statistical methods section

  • To understand the content of this study, this author should describe the hypothesis of this study.

We thank the reviewer for the request for clarification. Although an association between excess weight and elevated blood pressure values in children is well known and demonstrated, not all obese and overweight children are hypertensive. The hypothesis of the study is that insulin resistance may be one of the main mediators of the relationship between BMI (and waist circumference) and blood pressure. In our study, in addition to showing a mediating role exerted by insulin resistance in the relationship between BMI (and waist circumference) and blood pressure at baseline, we observe that changes in blood pressure associated with weight and waist circumference changes at follow-up seem to be mediated by the simultaneous modifications of insulin resistance. The child represents a unique model for studying the pathophysiology of essential hypertension, as confounding factors (aging, comorbidities, medications, smoking) are absent. Thus, our data could represent an important piece in the complex puzzle of the etiopathogenesis of essential hypertension. This concept is better clarified in the discussion of the revised manuscript.

  • Spell out the “HOMA” in this section.

The term HOMA was explained in full at the first citation

Material and method section

  • The range of age of present study population should be shown. In general clinical practice, BMI is used for participants with aged at and above 16 years while laurel Index is used for participants with aged between 5 to 15 years.

We thank the reviewer for this comment that allows us to clarify how we performed our analysis. We completely agree with the reviewer that BMI would not be an appropriate measure for children and adolescents aged < 16 years. In fact, we did not use this index but used the BMI z-score (see above). The age range of our population was 5.5-16.3 years. In most scientific papers dealing with pediatric populations in a similar age range, the BMI z-score is used as an indexed measure of weight. The BMI z-score is age- and sex-specific, and is used precisely so that individuals of different sexes and ages can be compared. All analyses in which this parameter is included are automatically adjusted for both sex and age. Further adjustment for the two parameters would result in over-correction.

  • The specific number of participants who are excluded from present study should be shown. The number of participants who has diabetes, secondary hypertension, taking antihypertension medication, and the participants who starts to taking antihypertensive medication during follow-up period should be clarified.

This information has been added in the methods section. We would like to point out that the number of children with secondary hypertension may seem low, but this is due to the fact that ours is a second level center for cardiovascular risk care, to which children aged >5-6 years are referred. The majority of secondary hypertension diagnoses (from nephropathy, endocrine pathology, and aortic coarctation) are usually made before this age. 

  • Multi-adjusted model which adjusted for known confounders should be added in present results. Especially for the influence of sex and age should be taken into consideration.  

As we explained above both age and sex were taken into account as both weight and blood pressure values were indexed for sex and age (z-score). Waist circumference was indexed for height (WtHr). In the revised manuscript, we added two multivariable analyses (new Table 2 and 4) to evaluate the association of HOMA index with systolic blood pressure, independently of BMI, WtHr and puberty. New Table 2 shows the uni- and multivariable model of the relationship between HOMA-Index and blood pressure at baseline, and new Table 4 shows the multivariable model of the effect of HOMA-index changes on blood pressure changes at follow-up.

Results section

  • As shown in Table 2, the correlation between SBP z-score and delta BMI (z-score) and the correlation between SBP z-score and delta HOMA index, showed wide range of 95% confidence intervals. Therefore, even p-value showed significant value, such results could indicate that the sample size of this population is too small to validate present results. In addition to that, this analyzes were performed by using crude model. Then validity of those results is questionable. At least, the results of good for fit test should be shown.

To address the reviewer's observation, we added a multivariable model with delta systolic blood pressure z-score at follow-up as outcome and delta HOMA index as predictor, adjusted by delta BMI z-score, delta WtHr and pre-pubescent to pubescent transition. The results are presented in new Table 4.

  • The abbreviation of HOMA should be explain in foot note

The abbreviation has been explained in the notes 

Discussion section

  • By using the multivariable model, the analyzes should be reperformed. And the content of discussion should be changed according to revised results.

Results regarding the multivariable analyses performed were embedded into the discussion

Reviewer 2 Report

In the manuscript, entitled “Relationship between body weight, waist circumference, insulin resistance and blood pressure in a pediatric population before and after a non-pharmacological treatment based on dietary behavioural modifications” submitted to Metabolites for a potential publication, the authors present their research work on relationship between anthropometric measurements, blood pressure and insulin resistance as well as the role of dietary modifications. I am of opinion, that in the present form, it is not good enough to be published in this journal. My comments are presented below.

My comments:

1.       In general, the area of research as well as the results are well known, therefore the authors have to clearly state the potential novelties of the present study.

2.       The instructions on recommended diet and physical activity have to be clearly stated and standardised to prevent biases.

3.       The role of HOMA index in diastolic blood pressure has to be commented.

4.       Multivariate analysis of factors involved in hypertension development would be of value.

5.       Some other markers of insulin resistance should be presented and compared to present ones.

6.       Almost half of the references are more than ten years old.

7.       Discussion section has to be written in more structured manner. Importantly, the results have to be compared to previously published studies.

Author Response

In the manuscript, entitled “Relationship between body weight, waist circumference, insulin resistance and blood pressure in a pediatric population before and after a non-pharmacological treatment based on dietary behavioural modifications” submitted to Metabolites for a potential publication, the authors present their research work on relationship between anthropometric measurements, blood pressure and insulin resistance as well as the role of dietary modifications. I am of opinion, that in the present form, it is not good enough to be published in this journal. My comments are presented below.

We thank the reviewer for his/ her updated and accurate comments that allowed us to improve the manuscript and gave us valuable suggestions for further studies to be performed in our population

My comments:

  1. In general, the area of research as well as the results are well known, therefore the authors have to clearly state the potential novelties of the present study.

Our study not only confirms the already known relationship between insulin resistance and blood pressure in children, but suggests something new. The novelty of the study is the idea that insulin resistance may be one of the main mediators of the relationship between BMI (and waist circumference) and blood pressure. In our study, in addition to showing a mediating role exerted by insulin resistance in the relationship between BMI (and waist circumference) and blood pressure at baseline, we observe that changes in blood pressure associated with weight and waist circumference changes at follow-up, seem to be mediated by the simultaneous modifications of insulin resistance. The child represents a unique model for studying the pathophysiology of essential hypertension, as confounding factors (aging, comorbidities, medications, smoking) are absent. Thus, our data could represent an important piece in the complex puzzle of the etiopathogenesis of essential hypertension. This concept is better clarified in the discussion of the revised manuscript.

  1. The instructions on recommended diet and physical activity have to be clearly stated and standardised to prevent biases.

      Our study is not randomized, as the parents of the young patients would not have signed consent to a randomized trial. The reason is that the children had been referred to our center for resolution of a clinical problem, and their parents would not have agreed that their children could be put in a control group. So, a control group is missing, and this is a limitation of the study and was highlighted in the revised version of the manuscript. However, the purpose of our study was not to show that our intervention reduced blood pressure in all children, but to see if, in patients in whom the intervention resulted in a reduction of weight (and of WtHr) and blood pressure values, this could be in part due to the reduction in insulin resistance. The BMI z-score, WtHr, HOMA-index, and SBP z-score were analyzed as continuous variables. In this way, both children in whom improvement in the parameters was observed at follow-up and those in whom this did not occur were included in the analysis. To avoid confusion on this point, we thought it would be appropriate to change the title of the manuscript as follows " Role of insulin resistance as a mediator of the relationship between body weight, waist circumference, and blood pressure in a pediatric population."

  1. The role of HOMA index in diastolic blood pressure has to be commented.

In our dataset, HOMA was not significantly associated with DBP z-score values at baseline. In addition, HOMA had no significant mediating role in the relationship between BMI z-score and WtHr and DBP z-score at baseline. Therefore, we decided not to present the follow-up diastolic blood pressure data (which in any case confirmed an absence of HOMA mediation), so as not to burden the manuscript. If the reviewer deems it necessary, we can present this information as supplementary material.

  1. Multivariate analysis of factors involved in hypertension development would be of value.

      By analyzing our data, we used BMI and blood pressure z-scores. The z-score is age- and sex-specific, and is used instead of the raw data in order to compare individuals of different sex and age with respect to a specific parameter. Further adjustment for sex and age would result, in our opinion, in over-correction. Since nomograms obtained from a large enough pediatric population to create z-scores were not available for waist circumference, waist circumference was indexed for height (WtHr), as suggested by several authors (Maffeis et al. Waist-to-Height Ratio: a useful index to identify high metabolic risk in overweight children. J Pediatr 2008;152:207-213; Garnett et al. Waist-to-Height Ratio, a simple option for determining excess central adiposity in youmg people. Int J Obes 2008; 32:1028–1030; Mokha JS et al. Utility of Waist-to-Height ratio in assessing the status of central obesity and related cardiometabolic risk profile among normal weight and overweight/obese children: The Bogalusa Heart Study.BMC Pediatrics 2010; 10:73).

As suggested by the reviewer, to better understand the relationship between HOMA index and blood pressure, two multivariable models were added to both the analysis of the data at baseline and the analysis of the data at follow-up. Both analyses confirmed a significant association between HOMA index and SBP z-score at baseline (Table 2) and between delta HOMA index and delta SBP z-score from baseline to follow-up (Table 4), after adjustment by BMI z-score and WtHr (analysis at baseline) and by BMI z-score, WtHr and transition from pre-puberty to puberty (analysis at follow-up).

  1. Some other markers of insulin resistance should be presented and compared to present ones.

      We thank the reviewer for the very useful suggestion. Several new indices of insulin resistance associated with the development of hypertension in adults have been suggested in recent years (Yuan Y, Sun W and Kong X (2022), Front. Cardiovasc. Med. 9:912197.doi: 10.3389/fcvm.2022.912197). Specifically, Triglyceride-glucose (TyG) index, calculated as Ln [fasting triglycerides (mg/dL) × fasting glucose (mg/dL)/2], is proposed to be a reliable and ideal surrogate of insulin resistance. Higher TyG index may be associated with higher odds of hypertension (Wang Y, Yang W and Jiang X (2021) Association Between Triglyceride-Glucose Index and Hypertension: A Meta-Analysis. Front. Cardiovasc. Med. 8:644035. doi: 10.3389/fcvm.2021.644035) in general adult population. In pediatric population data about new index are few, however there is some evidence that higher TyG idex is associated with elevated blood pressure also in children and adolescents (Eur J Pediatr. 2019 Jul;178(7):1069-1074. doi: 10.1007/s00431-019-03392-x. Epub 2019 May 12). It would be really interesting to test with a mediation analysis whether TyG index has a role in mediating the effect of BMI z-score and WtHr on blood pressure and to compare the role of mediation of TyG index to that of HOMA index in our population. However, we think this should be the subject of another study, which could also include other indices of insulin resistance identified for the adult population (Yuan Y, Sun W and Kong X (2022), Front. Cardiovasc. Med. 9:912197.doi: 10.3389/fcvm.2022.912197). If, however, the reviewer and editor consider this analysis essential in order to publish the manuscript, we can perform it, but we would need more time than we are currently allowed for review and it would considerably increase the length of the manuscript. In any case, we thank the reviewer again for the suggestion, which we promise to use without fail in our next study

  1. Almost half of the references are more than ten years old.

We apologize to the reviewer for this inaccuracy. The references have been updated

  1. Discussion section has to be written in more structured manner. Importantly, the results have to be compared to previously published studies.

The discussion has been restructured. However, given the novelty of our results (there are no mediation analysis studies on this topic in children in the literature), no comparison studies with ours are available. Our results therefore need to be confirmed by other studies.

Reviewer 3 Report

This paper addresses interesting and important points. It is well written and statistics are, as usually in the papers of this group, really appropriate.

While I'm perfectly fine with the conclusions of the authors, some concerns arise from this study, some minor and some major.

1) first of all the reference list is quite old and should be updated;

2) row 20, I suppose 18.6 (9.3) would represent average and SD values, but this should be specified, even in the abstract;

3) rows 79-80, This is redundant. Fair interaction among clinicians is part of the good clinical practices, and so it was implied;

5) rows 103-104, The OMRON 705 has been validated through the old ESH 2002 and this should be declared;

6) Dietary salt reduction on rows 137-138: 5 grams of salt is equivalent to 1.97 grams of sodium, so "near equivalent" to 2 grams seems better;

7) Dietary salt reduction on rows 168-169: again, salt reduction seems only proposed and not verified;

8) Table 1: height in cm = 1.46 to 1.54 -> this seems amazing;

9) Table 1: BP seems (and actually "is", at least from a statistical point of view) unchanged;

10) Table 1: Puberty is (of course) significantly different between initial and final observations, this has not been taken into account during the discussion;

11) What I find really odd is the insufficient analysis (if any) of the dietary sodium before, during and after the follow-up (actually the dietary sodium received no consideration). In my opinion this debases the value of the present data. Without an appropriate assessment of dietary sodium, the relationships among the parameters here shown cannot be corrected for the primary dietary factor able to influence BP.

Finally, simple relationships are not irrefutable evidences of cause-effect links.

Author Response

This paper addresses interesting and important points. It is well written and statistics are, as usually in the papers of this group, really appropriate.

While I'm perfectly fine with the conclusions of the authors, some concerns arise from this study, some minor and some major.

We thank the reviewer for his/her review that allowed us to improve our manuscript and point out its limitations

  • first of all the reference list is quite old and should be updated;

The reference list has been updated

  • row 20, I suppose 18.6 (9.3) would represent average and SD values, but this should be specified, even in the abstract;

Yes, the value in brackets represents the standard deviation. This was specified as follows "18.6 (standard deviation , SD 9.3) months"

  • rows 79-80, This is redundant. Fair interaction among clinicians is part of the good clinical practices, and so it was implied;

We agree with the reviewer. The sentence has been removed.

  • rows 103-104, The OMRON 705 has been validated through the old ESH 2002 and this should be declared;

The reviewer is right, however this device was also recommended by the 2016 ESH guidelines, as it is one of the few devices validated in children and is currently used for pediatric blood pressure measurements even in recent studies ( e.g., Lurbe E, Hypertension. 2022 Jun;79(6):1237-1246. doi:10.1161/HYPERTENSIONAHA.122.18993. Epub 2022 Mar 29).

  • Dietary salt reduction on rows 137-138: 5 grams of salt is equivalent to 1.97 grams of sodium, so "near equivalent" to 2 grams seems better;

We thank the reviewer for the correct observation. The sentence has been corrected.

  • Dietary salt reduction on rows 168-169: again, salt reduction seems only proposed and not verified;

The reviewer is right. We could not verify whether the prescription regarding the amount of salt intake was followed correctly. This point was added as a limitation of the study.

  • Table 1: height in cm = 1.46 to 1.54 -> this seems amazing;

We thank the reviewer for the correct observation. The unit of measurement has been corrected (meters instead of centimeters)

  • Table 1: BP seems (and actually "is", at least from a statistical point of view) unchanged;

Sorry, but we don't quite understand the reviewer's observation. BP decreases significantly between baseline and follow-up (SBP z-score from 1.06 to 0.7 (p<0.001) and DBP z-score to 0.53 to 0.33 (p<0.001). Perhaps the reviewer was referring to the value of HOMA. If so, the reviewer is right.

  • Table 1: Puberty is (of course) significantly different between initial and final observations, this has not been taken into account during the discussion;

We thank the reviewer for the important observation. It is well known that insulin resistance increases with puberty. Therefore, we adjusted both the regression analysis at follow-up and the mediation model of the effect of HOMA reduction on SBP reduction at follow-up for prepubescent to pubescent transition. The new analyses were added to the manuscript (new Table 4 for multiple regression model and Figure S1 for mediation analysis). The adjustment did not change the result regarding the role of HOMA when delta BMI z-score was included in the model, whereas it blunted the effect of HOMA in the model with delta WtHr. This suggests that in subjects who become pubertal, the reduction in insulin resistance due to weight loss presumably is less pronounced than in those who remain prepubertal or had already begun pubertal development at baseline.

  • What I find really odd is the insufficient analysis (if any) of the dietary sodium before, during and after the follow-up (actually the dietary sodium received no consideration). In my opinion this debases the value of the present data. Without an appropriate assessment of dietary sodium, the relationships among the parameters here shown cannot be corrected for the primary dietary factor able to influence BP.

We agree with the reviewer that this is a limitation of the study and we pointed this out in the revised version of the manuscript. However, the purpose of the study was not to demonstrate the effect of the low-sodium diet on blood pressure, but to suggest that the children in whom the intervention was associated with a reduction in HOMA index were those in whom the greatest benefits in terms of improved blood pressure values were shown. We can only speculate, without having any evidence, however, that these children were those in whom the low-sodium diet was most closely followed. A commentary sentence at this point has been added to the discussion and the bibliography has been updated in accordance.

  • Finally, simple relationships are not irrefutable evidences of cause-effect links.

We also agree with the reviewer on this point. Mediation analysis only allows one to say that a factor acts as a mediator on the effect of one variable on another variable. It does not allow one to say that it is a causal relationship. This point was emphasized in the conclusion.

Reviewer 4 Report

This is my review on the article entitled: Relationship Between Body Weight, Waist Circumference, Insulin Resistance and Blood Pressure in a Pediatric Population Before and After A Non-Pharmacological Treatment Based on Dietary Behavioral Modifications. 

 Authors aim to study children with cardiovascular risk factors, undergoing non-pharmacological treatment and evaluate their blood pressure based on the insulin resistance levels (HOMA-index). Authors proved that the HOMA-index could act as a mediator of BMI, waist circumference and Systolic BP. 

Introduction is presenting all the necessary information and data for the justification of the present study. Materials and methods are thoroughly described. Inclusion and exclusion criteria are presented. Line 89 of BASELINE AND FOLLOW-UP paragraph you mentioned that “If deemed necessary, reinforcement was given to dietary and lifestyle recommendations”. Authors should mention in parenthesis more precise when the intervention was performed. Statistical analysis is the proper one. 

Results are clearly presented with all the appropriate tables and figures. 

Author Response

This is my review on the article entitled: Relationship Between Body Weight, Waist Circumference, Insulin Resistance and Blood Pressure in a Pediatric Population Before and After A Non-Pharmacological Treatment Based on Dietary Behavioral Modifications. 

 Authors aim to study children with cardiovascular risk factors, undergoing non-pharmacological treatment and evaluate their blood pressure based on the insulin resistance levels (HOMA-index). Authors proved that the HOMA-index could act as a mediator of BMI, waist circumference and Systolic BP.

We thank the reviewer for his/her positive judgment of our work.

Introduction is presenting all the necessary information and data for the justification of the present study. Materials and methods are thoroughly described. Inclusion and exclusion criteria are presented.

Line 89 of BASELINE AND FOLLOW-UP paragraph you mentioned that “If deemed necessary, reinforcement was given to dietary and lifestyle recommendations”. Authors should mention in parenthesis more precise when the intervention was performed.

In the event that a worsening of weight and/or blood pressure was found during the follow-up visit compared with the previous visit, an interview was conducted by the staff (pediatrician, cardiologist, and nutritionist) to ascertain whether and what part of the planned intervention had been disregarded (non-adherence to the diet, lack of physical activity, excess sedentary lifestyle). In particular, a careful dietary history was repeated to see if the child had particular difficulties in accepting the proposed diet. In this case, modifications and substitutions were suggested, maintaining the total caloric intake and the balance between different macronutrients, but taking into account the patient's tastes and preferences. It was also checked whether there were any difficulties or errors on the part of the parents in the preparation of meals, in the dosage of foods given, and in the amount of the seasoning. If deemed necessary, more frequent follow-up visits were prescribed. Since this was not a randomized study, more could not be done. These details have been added to the revised manuscript.

Statistical analysis is the proper one. 

Results are clearly presented with all the appropriate tables and figures. 

Round 2

Reviewer 1 Report

Unfortunately, I thought using the Z score could not validate present results. To evaluate the weight and height valance among children, there are mainly three types of indexes. And this author should re-consider the reason why there are many indexes for children. The growing period influence on the balance between the length of trunk and legs. And the present study population showed comparative wide range of age. Even the age showed 10.70±2.46, the range of age could be 8.0 to 13.0 years old. Therefore, the BMI values for child of 8 years could not same to BMI values for children of 13 years. This is the reason why, Z-score could not validate present analysis. Well managed study with population with narrow range of age is mandatory for present analysis.

Author Response

Sorry, but we disagree with the reviewer's statement. Virtually, the entire pediatric world uses BMI z-scores to index weight in children two years of age and older. WHO recommends this, and the most recent guidelines on children obesity of the American Academy of Pediatrics (PEDIATRICS Volume 151, number 2, February 2023: e2022060640) write : " BMI is a measure used to screen for excess body adiposity; it is calculated by dividing a person’s weight in kilograms by the square of height in meters. For children and teens, BMI interpretation is age- and sex-specific. A child’s BMI category (eg, healthy weight, overweight) is determined using an age- and sex specific percentile for BMI rather than the BMI cut-points used for adult categories (ref 72)."Reference 72 of the American Academy of Pediatrics document refers to the CDC site, which was cited and used for our study as the reference population for calculating z-scores (Centers for Disease Control and Prevention. Body mass index (BMI). Available at: https://www.cdc.gov/ healthyweight/assessing/bmi/index).

The z-score is nothing more than a different way of expressing percentile. The Consensus of the American Academy of Pediatrics, on page 13 also writes: " Conversion of BMI percentiles to z-scores (a statistical measure that describes a value’s relationship to a population mean) derived from the CDC Growth Charts have historically been used for assessing longitudinal change in adiposity over time among children and adolescents with obesity".

These points were clarified in the revised version of the manuscript, and the American Academy of Pediatrics document was quoted.

Reviewer 2 Report

The revised version of the manuscript, entitled »Relationship between body weight, waist circumference, insulin resistance and blood pressure in a pediatric population before and after a non-pharmacological treatment based on dietary behavioral modifications«, and renamed »Role of insulin resistance as a mediator of the relationship between body weight, waist circumference, and systolic blood pressure in a pediatric population« submitted to Metabolites for a potential publication, has been considerably improved taking most of my suggestions into consideration. Anyway, I am of opinion, that it is still needs some minor revisions. Namely, the results regarding diastolic blood pressure have to be included in the manuscript. In addition, there are still some references needing renovation.

Author Response

As requested by the reviewer, results regarding diastolic blood pressure were added to the manuscript as two additional figures (Figure S1, panel a and b and Figure S3, panel a and b). In addition, 4 more recent literature references have been added.

Reviewer 3 Report

I appreciated the efforts of the authors to improve the quality of the manuscript. However, with no data about salt consumption (dietary sodium seems not considered in the study schedule), this paper seems devoid of value.

I suggest the authors to replan their study, including UNaV before and after their follow up.

Author Response

Unfortunately, we do not have urinary sodium excretion data. The effect of dietary salt restriction as a tool to reduce blood pressure is known and documented in adults [new ref 45] and suggested for pediatric populations [new ref 46]. However, performing a 24-hour urine sodium collection before and after our intervention period would not have been enough to give a measure of the actual dietary sodium intake of our patients, but would only have given information on the intake of salt the day preceding the test. To have consistent data on the level of adherence to the low sodium diet, we would need to measure urine sodium repeatedly and frequently throughout the follow-up period. This was not possible, given the size of the sample and the need not to ask too much of the children's families. It should also be emphasized that sodium intake estimation formulas derived from urinary sodium are not validated in children.

However, our study was not aimed at showing that low-sodium diet reduces blood pressure values, but to highlight how reductions in blood pressure (and even increases) go hand in hand with reductions (and increases) in insulin resistance, and this, in our opinion, is a new and important message, because it may help us understand why not all obese children are hypertensive.